# Physiological Roles of ERM Proteins and Transcriptional Regulators in Supporting Membrane Expression of Efflux Transporters as Factors of Drug Resistance in Cancer

**DOI:** 10.3390/cancers12113352

**Published:** 2020-11-12

**Authors:** Takuo Ogihara, Kenta Mizoi, Hiroki Kamioka, Kentaro Yano

**Affiliations:** 1Graduate School of Pharmaceutical Sciences, Takasaki University of Health and Welfare, 60, Nakaorui-machi, Takasaki, Gunma 370-0033, Japan; 1720101@takasaki-u.ac.jp; 2Faculty of Pharmacy, Takasaki University of Health and Welfare, 60, Nakaorui-machi, Takasaki, Gunma 370-0033, Japan; mizoi-k@takasaki-u.ac.jp (K.M.); kentarou.yano@yok.hamayaku.ac.jp (K.Y.); 3Laboratory of Drug Metabolism and Pharmacokinetics, Yokohama University of Pharmacy, 601, Matano-cho, Totsuka-ku, Yokohama, Kanagawa 245-0066, Japan

**Keywords:** efflux transporters, ERM proteins, transcriptional regulators, SNAI family, epithelial–mesenchymal transition, membrane expression, P-glycoprotein

## Abstract

**Simple Summary:**

Multidrug resistance and infiltration/metastasis of cancer are two major issues that must be overcome during chemotherapy. Efflux transporters, such as P-glycoprotein (P-gp), multidrug resistance-associated proteins, and breast cancer resistance protein play an important role in multidrug resistance of cancer. The ERM scaffold protein ezrin, radixin, and moesin acts as an anchor to support the expression of these transporters on the cell membrane. This review summarizes the organ-specific relationship between efflux transporters and ERM proteins. Furthermore, we also describe how transcriptional regulatory factors such as Snail that play an active role in cancer metastasis, are involved in the activation of P-gp through the expression of ERM proteins. We hope that this review will be useful to many researchers in understanding how to overcome multidrug resistance in cancer.

**Abstract:**

One factor contributing to the malignancy of cancer cells is the acquisition of drug resistance during chemotherapy via increased expression of efflux transporters, such as P-glycoprotein (P-gp), multidrug resistance-associated proteins (MRPs), and breast cancer resistance protein (BCRP). These transporters operate at the cell membrane, and are anchored in place by the scaffold proteins ezrin (Ezr), radixin (Rdx), and moesin (Msn) (ERM proteins), which regulate their functional activity. The identity of the regulatory scaffold protein(s) differs depending upon the transporter, and also upon the tissue in which it is expressed, even for the same transporter. Another factor contributing to malignancy is metastatic ability. Epithelial–mesenchymal transition (EMT) is the first step in the conversion of primary epithelial cells into mesenchymal cells that can be transported to other organs via the blood. The SNAI family, a transcriptional regulators triggers EMT, and SNAI expression is used is an indicator of malignancy. Furthermore, EMT has been suggested to be involved in drug resistance, since drug excretion from cancer cells is promoted during EMT. We showed recently that ERM proteins are induced by a member of the SNAI family, Snail. Here, we first review recent progress in research on the relationship between efflux transporters and scaffold proteins, including the question of tissue specificity. In the second part, we review the relationship between ERM scaffold proteins and the transcriptional regulatory factors that induce their expression.

## 1. Introduction

The World Health Organization (WHO) predicts that the incidence of cancer worldwide in 2020 will be 22 million, and the death toll will reach 13 million people. The malignancy of cancer greatly affects the 5-year survival rate of patients after the start of treatment, and factors that contribute to malignancy include the acquisition of multidrug resistance and metastasis/invasion ability. With regard to the acquisition of multidrug resistance, multiple mechanisms can reduce the sensitivity of cells to anticancer drugs, including increased resistance to apoptosis [1], suppression of oxidative stress [2], formation of microenvironments supportive of cancer cells [3], and increased expression of drug efflux transporters [4,5,6,7].

During the course of chemotherapy with anti-cancer drugs, cancer cells often acquire multidrug resistance via increased expression of efflux transporters such as P-glycoprotein (P-gp, ABCB1), multidrug resistance-associated proteins (MRPs, ABCCs), and breast cancer resistance protein (BCRP, ABCG2), thereby reducing the intracellular accumulation of anticancer drugs [8,9,10]. These transporters use energy from the hydrolysis of adenosine triphosphate (ATP) to drive efflux transport of physiological substrates, as well as xenobiotics such as etoposide, vinblastine, and 7-ethyl-10-hydroxycamptothecin (SN-38, the active metabolite of irinotecan) [11,12,13]. Recently, several molecular-targeted drugs, targeting proteins such as the fusion protein of echinoderm microtubule-associated protein-like 4 (EML4) with anaplastic lymphoma kinase (ALK) (EML4-ALK) and epidermal growth factor receptor tyrosine kinase (EGFR TK), have been launched as therapeutic drugs for lung cancer. However, these drugs are substrates of P-gp and breast cancer resistant protein (BCRP) [14,15,16,17]. These transporters are expressed not only in cancer cells, but also in normal tissues, such as the intestine, liver, kidney, brain, adrenal gland, and lung [18,19,20,21,22,23,24,25,26,27]. Probably because of the wide distribution in both cancerous and normal tissues and the physiological roles of these transporters, attempts to use inhibitors of efflux transporters clinically to improve the efficacy of anti-cancer drugs have been unsuccessful [28,29,30].

mRNA expression level is often used as an indicator of protein expression and function [31]. However, the mRNA expression levels of transporters do not necessarily correlate with protein expression level or functional activity [32], as exemplified by P-gp [33,34]. Transporters exhibit functional activity only when expressed on the cell membrane [33,35,36], and therefore scaffold proteins that anchor the transporter to the cell membrane play a critical role. Recently, several research groups including our own have reported that ezrin (Ezr), radixin (Rdx), and moesin (Msn), which are generically known as ERM proteins, serve as anchors to immobilize a variety of proteins, including transporters, at the cell membrane. Thus, it is suggested that the expression levels of ERM proteins regulate the functional activity of efflux transporters. But, the relevant regulatory scaffold protein(s) differs depending upon the transporter, and also upon the tissue in which it is expressed, even for the same transporter [37]. In the first half of this review, we will focus on the relationship between efflux transporters and scaffold proteins, including the tissue specificity.

Another important factor influencing the malignancy of cancer is metastatic ability. When cancer cells metastasize, epithelial–mesenchymal transition (EMT) is the first step in the conversion of primary epithelial cells into mesenchymal cells, which can be transported in the blood. Cancer cells lose their polarity and cell adhesiveness during EMT, and the resulting mesenchymal cells acquire the ability to migrate and invade [38,39]. Members of the SNAI family (Snail (SNAI1), Slug (SNAI2), and Smuc (SNAI3)) of transcriptional regulators trigger EMT, and are considered useful indicators of the malignancy of cancer. EMT has also been suggested to be involved in drug resistance, since drug excretion from cancer cells is promoted during EMT, though the mechanisms involved have not yet been fully clarified. We recently showed that Snail induces ERM proteins [36]. In the second part of this article, we will review recent research, including clinical findings, on the relationship between the ERM scaffold proteins and the transcriptional regulatory factors that induce their expression.

## 2. Relationship between ERM Scaffold Proteins and Efflux Transporter P-gp

P-Glycoprotein (P-gp) was first identified as an efflux transporter associated with multidrug resistance (MDR) in cancer cells such as adenocarcinoma and leukemia [19,40]. It is now known that this efflux transporter is expressed not only in tumor tissues, but also in normal tissues such as the intestine, [29,41] liver, [42] kidney, [25] brain [43,44], and adrenal gland [45]. P-gp serves to protect these organs by mediating the efflux transport of xenobiotics. In particular, P-gp is thought to act as a barrier to drug absorption in the intestine and also to drug distribution to the brain, as well as tumor cells. Therefore, P-gp function, gene expression, and substrate specificity have been extensively investigated to support drug development [37].

ERM proteins are located at cell membranes in many tissues, including the liver, digestive tract, kidney and lung [46]. The N-terminal of ERM proteins is composed of a domain of about 200 amino acids that is called the four-point-one Ezr, Rdx, and Msn (FERM) region. ERMs also have an α-helix region in the central region, and an F-actin-binding domain at the C-terminal, consisting of about 100 amino acids. ERM proteins show more than 75% homology with each other [47]. ERM proteins are inactive when the C-terminal and N-terminal interact intramolecularly, and cannot bind to actin [48,49,50]. However, when the C-terminal tyrosine is phosphorylated, these proteins bind to actin filaments and are fixed on the cell membrane [51,52,53,54,55]. This phosphorylation of ERM occurs upon activation of RhoA, one of the small GTP-binding proteins, through a pathway involving phosphatidylinositol 4-phosphate 5-kinase (IPI4P5K) and its product phosphatidylinositol 4,5-bisphosphate (PIP2) [56]. The general role of the ERM proteins is to cross-link between proteins located on plasma membranes and cortical actin filaments. ERM proteins bind to integral membrane proteins, scaffold proteins, and Rho-related proteins (Table 1) [57].

It was recently established that ERM proteins act as scaffolds to anchor efflux transporters to the cell membrane, thereby regulating the functional activity [36]. Luciani et al. found that treatment of macrophages with interferon γ increased the expression of P-gp on the cell membrane and its co-localization with Ezr among ERM proteins. They also found that P-gp binds to Ezr, Rdx, and Msn and is anchored to actin through this binding in a study using vinblastine-resistant T-cell lymphoma cells [58]. Furthermore, it was shown that inhibition of the action of ERMs with antisense oligonucleotides broadened the distribution of P-gp from the cell membrane to the whole cell. These findings indicate that the intramembrane expression of P-gp is regulated by ERM. Moreover, Brambilla et al. investigated the binding sites of P-gp and Ezr in osteosarcoma [59]. They showed that the FERM domain (amino acid residues 149–242) of Ezr was required for binding to P-gp, and mutation in this region reduced drug tolerance.

Among the ERM proteins, Rdx is involved in the localization of P-gp on the cell membrane of hepatocytes. Wang et al. monitored the localization of P-gp in the plasma membrane of rat hepatocytes after knockdown of Rdx with RNAi. In the Rdx-knockdown cells, P-gp was not localized on the membrane surface, but was internalized in the cytoplasm [60]. We investigated the effect of knockdown of ERM proteins on P-gp expression and activity at the transcriptional, translational, and post-translational levels, using liver HepG2 as a model cell line. Knockdown of Ezr with RNAi reduced the level of P-gp mRNA, and the levels of P-gp protein in the whole cell and on the cell surface compared to the control, but without statistical significance [61]. Knockdown of Msn caused no change in the expression of P-gp mRNA or in the protein amount or distribution of P-gp. On the other hand, the knockdown of Rdx significantly decreased P-gp protein expression on the cell surface to 30% compared with control cells, without significantly altering P-gp mRNA or protein expression in the whole cell. Moreover, a significant increase in the accumulation of Rhodamine 123 (Rho123), a typical P-gp substrate, was observed in Rdx-knockdown cells, suggesting reduced functional activity of P-gp. These results indicate that Rdx is mainly involved in the membrane localization of P-gp at the translational level in hepatic cells. Examination of Rho123 accumulation in cells with knockdown of various combinations of ERM proteins (Ezr/Rdx, Rdx/Msn, Ezr/Msn, and Ezr/Msn/Rdx) revealed additive, but not synergistic, effects on P-gp activity.

We also investigated the role of ERM proteins in the regulation of P-gp transport activity in cancer cells, using human colon adenocarcinoma (Caco-2) cells and renal cancer (Caki-1) cells [62]. When each ERM protein was knocked down with siRNA, the mRNA level of P-gp remained unchanged in Caco-2 cells. However, the P-gp-mediated intracellular accumulation of Rho123 was increased when Rdx, but not Ezr or Msn, was silenced. In contrast, P-gp activity in Caki-1 cells was not affected by knockdown of any of the ERM proteins. We also performed the physiological role of Rdx in regulating P-gp expression and activity in the small intestine and kidney using Rdx and P-gp knockout mice [35]. In intestinal tissue homogenates of Rdx-knockout mice, P-gp protein levels were increased from the upper to the lower intestine in both wild-type and Rdx−/−mice, but there was no difference in expression levels between the two groups. On the other hand, in the cell membrane fraction, a tendency for increase of P-gp was observed from the upper part of the small intestine to the lower part in the wild-type mice, whereas the level was the same from the upper part to the lower part of the small intestine in Rdx−/−mice. When Rho123 was orally administered to Rdx−/− and wild-type mice, the plasma concentration of Rho123 in the absorption phase was higher in Rdx−/− than in wild-type mice, but there was no difference in the half-life of the elimination phase between the two groups. On the other hand, when Rho123 was orally administered to P-gp knockout (mdr1a/b−/−) and corresponding wild-type mice, the disappearance of Rho123 was suppressed only in mdr1a/b−/− mice. These findings suggest that Rdx functions as an anchor protein for gastrointestinal P-gp, which influences the absorption phase of Rho123, but not for renal P-gp, which affects the elimination phase of Rho123. These findings are at least consistent with the results of in vitro studies using corresponding cancer cells, although there is still debate about the species difference between mouse and human. That is, it appears that the nature of the regulation of P-gp by ERM proteins is tissue-dependent, but is similar in cancer and normal cells of a given tissue. Tokuyama et al. reported that repeated oral administration of the P-gp substrate drug etoposide to mice activated Rdx in intestinal epithelial cells and increased the expression of P-gp on the membrane [63]. They observed a decreased analgesic effect of morphine, a P-gp substrate drug, compared to etoposide-untreated mice, indicating that etoposide increased the membrane expression and transport function of P-gp through activation of Rdx in the small intestine. Based on this and our findings, the existence of a relationship between P-gp and Rdx in the digestive tract has been widely accepted.

It has been reported that ERM activation is mediated by RhoA signaling [63]. This is consistent with the report by Tsukita et al. [56]. They showed that RhoA simultaneously activates Rho kinase (ROCK) and PI4P5K (which generates PIP2), and that PIP2, not ROCK, is involved in ERM phosphorylation. On the other hand, Kobori et al. indicated that RhoA phosphorylates ERM through activation of ROCK [64]. Kobori et al. used mouse small intestine, whereas Tsukita et al. overexpressed RhoA in mouse fibroblast 3T3 cells and human cervical cancer-derived HeLa cells. Therefore, the difference between the experimental systems is notable, and the results cannot be directly compared. In any case, it is considered that a phosphorylation signal is required for the activation of ERM proteins.

It is not yet clear how P-gp substrates activate RhoA and other factors. We found that a bitter substrate of P-gp such as etoposide activates ERM and enhances the membrane expression of P-gp via stimulation of cholecystokinin receptor. Since this stimulation causes activation of RhoA [65], it is possible that etoposide also stimulates the cholecystokinin receptor through stimulation of the bitter taste receptor and causes phosphorylation of ERM proteins. Therefore, it seems plausible that during continuous administration of anticancer drugs, the intramembrane expression level of P-gp is increased through activation of ERMs by the drugs via the above pathways, and as a result, the transport function is enhanced.

We also investigated whether ERM proteins functionally regulate P-gp in non-small cell lung cancer HCC827 cells [36]. In HCC827 cells, Ezr or Msn knockdown significantly reduced the Rho123 efflux rate. These results suggest that Ezr or Msn is involved in membrane expression of P-gp in the lung.

P-gp is also expressed in cerebrovascular endothelial cells and is a component of the blood–brain barrier (BBB). Kobori et al. found that P-gp and Msn were co-expressed on the cell membrane [66]. Moreover Terasaki et al. knocked down ERM proteins in human cell hCMC/D3, which is considered to be a model of the BBB [67]. They found that knockdown of Rdx decreased the expression of P-gp on the cell membrane, while knockdown of Msn also decreased the activity of P-gp. These reports suggest that the P-gp scaffolding protein at the BBB may be Rdx or Msn, or both. Moreover, Zhang et al. showed that P-gp and actin form a complex with Ezr in mouse brain capillary endothelial cells, suggesting that Ezr also regulates P-gp [68]. The apparent discrepancies among these reports might be due at least in part to the difference of animal species, but the details remain unclear. However, we can at least say that one or more of the ERM proteins regulates P-gp transport function in brain capillary endothelial cells.

Overall, findings to date indicate that the regulation of P-gp function by ERM proteins is organ-specific (Table 2). Therefore, it may be possible to increase the accumulation of P-gp substrate drugs in lung, intestinal or renal cancer tissues in a tissue-specific manner by targeting the appropriate ERM. The reason for the tissue specificity has not been established, but it may be related to the high expression levels of Rdx in the liver and of Msn in the vascular endothelium [69,70,71].

## 3. Relationship between ERM Scaffold Proteins and Efflux Transporter MRP2

MRPs are localized to the apical and/or basolateral membrane of hepatocytes, enterocytes, renal proximal tubule cells, and endothelial cells of the blood–brain barrier. Most human MRPs can transport organic anion drugs conjugated to glutathione, sulfate, or glucuronic acid. In addition, anti-cancer agents and their metabolites, platinum compounds, folic acid antimetabolites, nucleosides and nucleotide analogs are substrates of MRPs. It is important to note that MRP2 transports the glucuronide conjugate of bilirubin as an endogenous substrate and is involved in the excretion of this substance from the liver through the bile duct. Therefore, inhibition of MRP2 can cause hepatotoxicity. Moreover, MRP 1 to 3 are involved in multidrug resistance of tumors and contribute to the extracellular efflux of many anticancer drugs and the reduction of their intracellular accumulation. Overcoming MRPs-mediated multidrug resistance is a key issue in cancer chemotherapy [79].

Among the transporters expressed in the apical membrane of the bile duct of the liver, MRP2 (ABCC2) is involved in the biliary excretion of drugs such as pravastatin [80], indomethacin glucuronide [81], and SN-38 glucuronide [82]. Several researchers have proposed that Rdx regulates MRP2 function in the liver [83]. For example, Rdx-deficient mice develop hyperbilirubinemia due to loss of MRP2 membrane expression. Oxidative stress inhibits the interaction between MRP2 and Rdx, causing MRP2 to be internalized in the cell [72]. At this time, it is considered that reduced phosphorylation of threonine at the C-terminal of Rdx impairs the binding of MRP2 to actin, resulting in a decrease in the localization of MRP2 to the cell membrane. Clinically, changes in the expression and localization of MRP2 and Rdx have been observed in liver biopsy samples of patients with primary biliary cirrhosis [73]. Other groups obtained similar results for cell lines derived from hepatoma [74,75]. Iwaki et al. investigated the effect of knockdown of ERM proteins on the transport activity of MRP2 and the intracellular accumulation and efficacy of methotrexate (MTX), a substrate of MRP2, in liver cancer cell lines [74]. Rdx knockdown in human liver cancer HepG2 cells reduced the localization of MRP2 to the cell membrane and the extracellular transport activity of MTX. This reduction was blocked by treatment with siRdx, resulting in increased cytotoxicity of MTX. In an in vivo study, it was found that treatment of cancer-bearing mice with siRdx improved the pharmacological effect of MTX. All these studies support the idea that Rdx serves as an anchor protein that contributes to the cell membrane expression of MRP2 in the liver, both in normal tissues and in cancer cells.

The results of knockdown of ERM proteins in Caco-2 cells indicated that Rdx and/or Ezr is involved in the expression of MRP2 in the apical membrane of the gastrointestinal tract [78]. Sugiyama et al. established a stable Caco-2 cell line with knockdown of Rdx or Ezn by the RNAi method. MRP2 was not expressed on the cell surface of this cell line. This result was also supported by an immunoprecipitation assay. These findings suggest that both Rdx and Ezn are independently required for apical localization of MRP2 in the digestive tract. Studies using Caco-2 cells and rat small intestine also suggested that MRP2 membrane expression and transport function in the small intestine depend on the level of phosphorylated Ezr [76,77]. The earlier report by Sugiyama et al. [78] did not examine the phosphorylation level of Ezr, but knockdown of Ezr would also reduce the expression level of phosphorylated Ezr. Therefore, it seems that these reports are consistent. Furthermore, the transport function of MRP2 is regulated by Rdx in both the lung cancer-derived cell line A549 and the breast cancer-derived cell line MDA-MB-453 [74]. To summarize the case of MRP2 as shown in Table 2, the scaffold protein is often Rdx, but it seems to be tissue-specific as in the case of P-gp.

## 4. Relationship between ERM Scaffold Proteins and Efflux Transporter BCRP

Breast cancer resistance protein (BCRP, ABCG2) was first cloned from a breast cancer cell line resistant to chemotherapeutic agents. The substrate specificity of BCRP is extremely broad, and includes not only therapeutic agents such as SN-38, but also endogenous substances such as estrone-3-sulfate, 17β-estradiol17- (β-D-glucuronide), and uric acid. BCRP is also highly expressed in normal tissues such as syncytiotrophobic membrane, intestinal epithelium, hepatocytes, endothelial cells of cerebral microvessels, and apical membrane of renal proximal tubule cells. It is involved in the absorption, distribution, and excretion of many drugs and endogenous compounds, thereby regulating tissue exposure to these agents [84].

We also investigated whether ERMs functionally regulate BCRP in cell lines derived from lung, intestinal, and renal cancers [36]. BCRP functions were evaluated by means of excretion and uptake assays using non-small cell lung cancer HCC827 cells, colon cancer Caco-2 cells, and renal cancer Caki-1 cells. In HCC827 cells, efflux of SN-38, a substrate of BCRP, was significantly reduced by Ezr or Msn knockdown. Ezr or Rdx knockdown did not affect the function of BCRP in Caco-2 cells. The effect of Msn knockdown was not evaluated because Msn expression is extremely low in Caco-2 cells. In Caki-1 cells, Rdx knockdown increased intracellular SN-38 concentration, while Ezr or Msn knockdown had no effect. Moreover, Terasaki et al. suggested that the BCRP scaffold protein at the BBB [66] may be Rdx or Msn, as in the case of P-gp. These results indicate that the regulation of BCRP function by ERM is organ-specific, as in the case of P-gp (Table 2).

The membrane expression and transport function of BCRP are regulated by postsynaptic density 95/disc-large/zona occludens (PDZ) domain-containing protein, NHERF3 (also known as PDZK1, CAP70, or NaPi-Cap1) [85]. NHERF3 is a transporter adapter protein that regulates several transporters mediating influx of xenobiotics and nutrients in the small intestine. Western blot analysis of the intestinal brush border membrane revealed that BCRP expression was reduced in nherf3 (−/−) mice. This result was also supported by a decrease in apical localization of BCRP in nherf3 (−/−) mice, as assessed by immunohistochemical analysis. Transcellular transport of cimetidine, a substrate of BCRP [86], from the basement membrane to the apical membrane in MDCKII cells derived from canine kidney stably expressing both BCRP and PDZK1 is higher than that in cells stably expressing BCRP alone. In addition, the cells showed increased resistance to the cytotoxicity of SN-38. These results indicate that PDZK1 plays a crucial role in the apical localization of BCRP [85]. Na^+^/H^+^ exchanger regulators (NHERFs) are involved in immobilizing membrane proteins such as ion channels and receptors on the actin cytoskeleton through binding to ERM proteins. To date, three types of NHERFs are known, among which NHERF-1 forms a complex with the NHERF-2C terminal peptide and binds to the FERM domain of the ERM protein [87,88]. The relationship among NHERF-3, ERM proteins and BCRP is not yet known, but it is presumed that NHERF-3 also forms a complex with ERMs and contributes to the membrane expression of BCRP. NHERFs might also be involved in the membrane expression of other transporters, including influx transporters, besides BCRP. The factors that control the membrane expression of transporters seem to be considerably more complex than they had initially appeared.

The relationships between the efflux transporters and their scaffold proteins in different organs are summarized in Table 1. It seems clear that the expression levels of ERM proteins regulate efflux transporter activity. The regulatory scaffold protein(s) differs depending on the transporter, and also upon the tissue in which the transporter is expressed. Interestingly, there seems to be little difference between cancer cells and normal cells in the same tissue.

## 5. Change of P-gp Activity Associated with EMT

There are many reports showing that EMT is the first step in the acquisition of infiltration and metastasis capability by various cancer cells [89,90,91,92]. In EMT, the transforming cells lose apical polarity, acquire a mesenchymal phenotype, and become highly invasive and malignant. Snail, a member of the SNAI family of transcriptional regulators, triggers EMT, converting epithelial cells to mesenchymal cells that have migration and invasion ability [38,39]. Snail recognizes the E-box site on the target genome sequence and suppresses the transcription of genes such as the E-cadherin gene [93]. Snails are EMT [92,94] master regulators that promote cancer infiltration and metastasis [95,96,97]. For example, in lung cancer patients, high levels of Snail expression correlate with poor prognosis [98]. Moreover, patients with high P-gp expression levels also have a poor prognosis [99]. Thus, it is possible that Snail is involved in the expression and activity of P-gp. We investigated the involvement of ERM proteins in P-gp activation during Snail-induced EMT in lung cancer cells [100,101]. The mRNA expression levels of epithelial markers such as E-cadherin, occludin, and claudin-1 were reduced in Snail-overexpressing cells, whereas these of mesenchymal markers such as vimentin and ZEB1 were increased in Snail-overexpressing cells compared with Mock cells. Overexpression of Snail induced an increase of P-gp activity in HCC827 lung cancer cells concomitantly with increased cellular efflux of Rho123 and increased resistance to paclitaxel. At the same time, the mRNA expression level of Msn, but not Ezr or Rdx, was significantly increased, whereas the mRNA and protein expression levels of P-gp in whole cells were similar in Mock and Snail-overexpressing cells (Figure 1). The increase of P-gp activity was suppressed by knockdown of Msn.

Therefore, Msn mainly appears to regulate the intracellular localization of P-gp in lung cancer cells, resulting in an increase of P-gp activity during Snail-induced EMT. Several reports indicate that ERM proteins are related to cancer infiltration and metastasis, since the expression of these proteins is upregulated. For example, the silencing of Ezr expression suppressed cell polarization and invasive behavior in acute T lymphocytic cell line MOLT4 [102]. Overall, these reports on EMT suggest that ERM proteins might be promising targets for inhibiting the development of drug resistance, invasion and metastasis in cancer patients.

Most recently, we reported that the membrane localization and activity of MRP5, but not MRP2, were increased by Snail-induced EMT in HCC827 cells, resulting in the enhancement of cisplatin resistance. On the other hand, Snail overexpression significantly reduced protein expression levels of BCRP in both the homogenate and the plasma membrane fraction in HCC827 cells. As a result, BCRP function and resistance to SN-38 were decreased [103]. Thus, Snail does not seem to increase the activity of all efflux transporters. This information may prove useful in drug selection in chemotherapy for metastatic cancer.

We also examined clinical samples collected from lung cancer patients [101], and surprisingly found that the mRNA expression level of Snail was positively correlated not only with Msn, but also with Rdx and Ezr, as well as with P-gp mRNA. In other words, Snail appears to be indirectly involved in the increase of P-gp activity in lung cancer by stimulating the expression of Msn, according to in vitro studies. Moreover, EMT-inducing transcriptional regulators except Snail could be involved in increasing mRNA levels of Rdx and Ezr, and could also be directly involved in increasing P-gp mRNA and total intracellular expression of P-gp protein. However, the in vitro studies cannot fully explain the clinically observed phenomena, and the potential roles of other members of the SNAI family and other transcriptional regulators remain to be explored. It is also unclear whether the inductions of EMT and scaffold protein expression caused by Snail are independent or linked phenomena.

## 6. Conclusions

Several P-gp inhibitors have been developed with the aim of suppressing multidrug resistance of cancer cells, but none of them has been clinically applied, because of the problem of side effects arising from increased drug accumulation in normal tissues in addition to tumor tissues. However, since the scaffolding proteins involved in the membrane localization of P-gp are tissue-specific, control of the expression of the appropriate scaffolding protein might be a suitable strategy for achieving tissue-specific P-gp inhibition to overcome cancer resistance with reduced side effects. For example, colon cancer metastasizes to the lung via the bloodstream, but the scaffolding proteins of P-gp are different in the lung (Ezr or Msn) and digestive tract (Rdx). If the metastatic lesion inherits the characteristics of the primary lesion, and if the function of Rdx alone could be inhibited, it might be possible to specifically inhibit the P-gp function of cancer cells that have metastasized from the gastrointestinal tract to the lung without inhibiting P-gp in normal lung tissue. Furthermore, if the expression of transcriptional regulatory factors located upstream of scaffold protein expression could be controlled, it might be possible to create a drug that can simultaneously control not only drug resistance, but also cancer metastasis (Figure 2).

Nevertheless, at present this remains a very challenging goal. A great deal of work remains to be done to uncover the physiological mechanisms through which ERM proteins and transcriptional regulators regulate membrane expression of efflux transporters. In particular, multiple mechanisms, besides the increase of P-gp activity, are involved in the acquisition of drug resistance in lung cancer cells. We hope this review throws light on one possible approach to overcoming multidrug resistance in cancer.

## Figures and Tables

**Figure 1 cancers-12-03352-f001:**
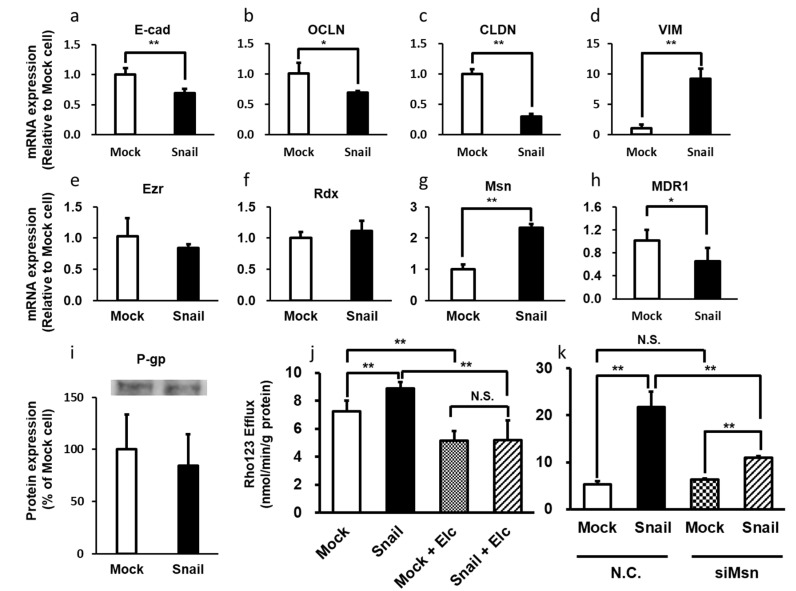
Snail-induced Epithelial-to-mesenchymal transition enhances P-gp-mediated multidrug resistance in HCC827 Cells. (**a**–**d**) The mRNA expression levels of epithelial markers E-cadherin, occludin, and claudin-1 were reduced in Snail-overexpressing cells. On the other hand, the mRNA expression levels of mesenchymal markers vimentin were increased in Snail-overexpressing cells relative to Mock cells, respectively. (**e**–**g**) The mRNA expression level of Msn was significantly increased in Snail-overexpressing HCC827 cells compared to Mock cells, but the expression levels of Ezr and Rdx were unchanged. (**h**) The mRNA expression level of MDR1 was decreased in Snail-overexpressing HCC827 cells compared to Mock cells. (**i**) There was no significant difference in P-gp protein expression levels between 2 groups. (**j**) The efflux rate of Rho123 was increased in Snail-overexpressing cells compared to Mock cells. Moreover, Rho123 efflux was inhibited by coincubation with P-gp inhibitor elacridar(Elc). (**k**) In Snail-overexpressing cells, the efflux rate of 10 mM Rho 123 was significantly increased compared to Mock cells, and this increase was significantly suppressed by knockdown of Msn. *: *p* < 0.05, **: *p* < 0.01, N.S.: not significant (Student’s *t*-test (**a**–**i**) and Holm test (**j**,**k**)).

**Figure 2 cancers-12-03352-f002:**
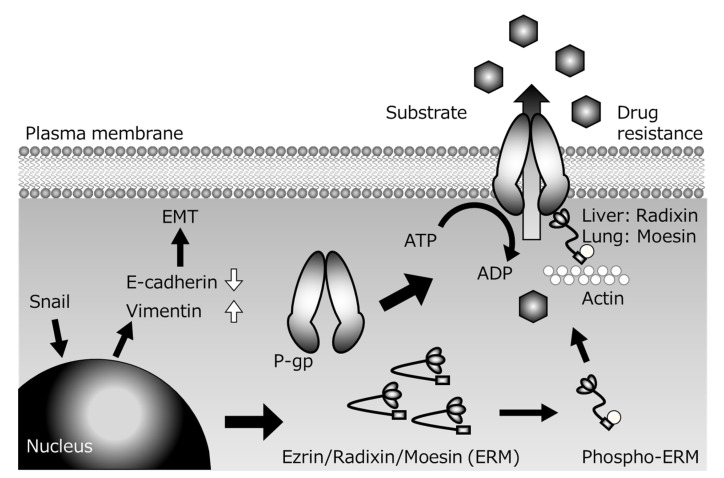
Presumed mechanisms of P-glycoprotein (P-gp)-mediated drug resistance during Snail-induce epithelial–mesenchymal transition (EMT).

**Table 1 cancers-12-03352-t001:** Proteins which bind to the FERM domains.

Membrane proteins
CD44
CD43
CD95 (APO-1/Fas)
Intercellular adhesion molecule-1 (ICAM-1)
Intercellular adhesion molecule-1 (ICAM-2)
L-selectin
MRP2
Na^+^,H^+^-exchanger (NHE1)
Na^+^ K^+^ 2Cl-cotransporter (NKCC2)
P-gp
Scaffoerd proteins
NHERF1 (EBP-50)
NHERF2
NHERF3
Rho-relate3 proteins
Dbl
Rho-GDP-dissociation inhibitor (Rho-GDI)

This table is modified from Table 1 quoted in the paper [57]. Reproduced in part with permission from Biol. Pharm. Bull. Vol. 40 No. 4. Copyright 2017, the Pharmaceutical Society of Japan.

**Table 2 cancers-12-03352-t002:** Efflux transporters and their presumed scaffold proteins.

Tissues	P-gp	MRP2	BCRP
Liver	Normal	Rdx [60]	Rdx [72,73]	
Cancer cell lines	Rdx [61]	Rdx [74,75]	
Intestine	Normal	Rdx [35]	Ezr [76,77]	
Cancer cell lines	Rdx [62]	Rdx [78], Ezr [78]	Not Ezr, Rdx [36]
Kidney	Normal	Not Rdx [35]		
Cancer cell lines	Not ERM [62]		Rdx [36]
Lung	Cancer cell lines	Ezr, Msn [36]	Rdx [74]	Ezr, Msn [36]
BBB	Normal	Msn [66], Ezr [68]		
Cancer cell lines	Msn, Rdx [67]		Msn, Rdx [67]
Breast	Cancer cell lines		Rdx [74]

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
