# Peer review of "Physiological Roles of ERM Proteins and Transcriptional Regulators in Supporting Membrane Expression of Efflux Transporters as Factors of Drug Resistance in Cancer"

_cancers, 2020, doi:10.3390/cancers12113352_

Round 1
Reviewer 1 Report
The paper provides a review of the role of ERM proteins in the regulation of activity and cellular localization of ABC-family membrane transporters. The role of ERM proteins in the regulation of the activity of ABC proteins is not a very popular research topic, therefore the publication gives the impression that the authors do not go too deep into the description of the mechanisms. On the other hand, the work may inspire MDR researchers to focus their attention on the role of ERM proteins in the process of drug resistance. The work is written very clearly and it is easy to read. I believe the paper is a valuable source of knowledge and may be published in Cancers.
Author Response
Thank you for your input. Please refer to the opinions of other reviewers.
Reviewer 2 Report
The manuscript „Physiological Roles of ERM Proteins and Transcriptional Regulators in Supporting Membrane Expression of Efflux Transporters as Factors of Drug Resistance in Cancer” describes the role of three proteins, ezrin (Ezr), radixin (Rdx) and moesin (Msn) in function of ABC transporters. The other point of the manuscript is concentrated on desciption of contribution of EMT into P-gp activity. The manuscript is well written. I propose to include description of EZR proteins, ezrin, radixin and moesion in body text. If it will be possible the authors could summarize the role ERM in table.
Author Response
Thank you for your comment. Following your suggestions, we have added several sentences, one reference and tables, so please check them.
P3L22-P4L7
The general role of the ERM proteins is to cross-link between proteins located on plasma membranes and cortical actin filaments. ERM proteins bind to integral membrane proteins, scaffold proteins, and Rho-related proteins (Table 1) [58].
Table 1 Proteins which bind to the FERM domains
|
Membrane proteins |
|
|
|
CD44 |
|
|
CD43 |
|
|
CD95 (APO-1/Fas) |
|
|
Intercellular adhesion molecule-1 (ICAM-1) |
|
|
Intercellular adhesion molecule-1 (ICAM-2) |
|
|
L-selectin |
|
|
MRP2 |
|
|
Na+,H+-exchanger (NHE1) |
|
|
Na+ K+ 2Cl-cotransporter (NKCC2) |
|
|
P-gp |
|
Scaffoerd proteins |
|
|
|
NHERF1 (EBP-50) |
|
|
NHERF2 |
|
|
NHERF3 |
|
Rho-relate3 proteins |
|
|
|
Dbl |
|
|
Rho-GDP-dissociation inhibitor (Rho-GDI) |
This table is modified from Table 1 quoted in the paper [58].
P13L23-24
- Kawaguchi, K.; Yoshida, S.; Hatano, R.; Asano, S.; Pathophysiological Roles of Ezrin/Radixin/Moesin Proteins. Biol. Pharm. Bull. 2017, 40, 381-390.
Reviewer 3 Report
In this review article, the authors summarized recent findings of the role of ERM proteins and transcriptional regulators in cancer. In particular, they focused on the relationship between efflux transporters and scaffold proteins, and between ERM scaffold proteins and the transcriptional regulatory factors induce their expression. This review brings together a wide body of information that may be of use for researchers interested in this field. My specific comment for this manuscript is listed below.
1. In Table 1, the column name “Transporter” should be “Organs” or “Tissues”. It will help readers to understand their meanings at a glance.
Author Response
Thank you for your advice. We have amended the Table 1, so please check it.